# Micro-sized open-source and low-cost GPS loggers below 1 g minimise the impact on animals while collecting thousands of fixes

**Timm A. Wild**[1,2,3]*, **Jens C. Koblitz**[1,2,4], **Dina K. N. Dechmann**[1,2], **Christian Dietz**[5], **Mirko Meboldt**[3], **Martin Wikelski**[1,2,4]

**1** Department of Migration, Max Planck Institute of Animal Behavior, Radolfzell, Germany, **2** Department of Biology, University of Konstanz, Konstanz, Germany, **3** Product Development Group Zurich (pd|z), ETH Zürich, Zürich, Switzerland, **4** Centre for the Advanced Study of Collective Behaviour, University of Konstanz, Konstanz, Germany, **5** Biologische Gutachten Dietz, Haigerloch, Germany

\* twild@ab.mpg.de

## Abstract

GPS-enabled loggers have been proven as valuable tools for monitoring and understanding animal movement, behaviour and ecology. While the importance of recording accurate location estimates is well established, deployment on many, especially small species, has been limited by logger mass and cost. We developed an open-source and low-cost 0.65 g GPS logger with a simple smartphone-compatible user interface, that can record more than 10,000 GPS fixes on a single 30 mAh battery charge (resulting mass including battery: 1.3 g). This low-budget 'TickTag' (currently 32 USD) allows scientists to scale-up studies while becoming a 'wearable' for larger animals and simultaneously enabling high-definition studies on small animals. Tests on two different species (domestic dog, *Canis lupus familiaris* and greater mouse-eared bats, *Myotis myotis*) showed that our combination of optimised hardware design and software-based recording strategies increases the number of achievable GPS fixes per g device mass compared to existing micro-sized solutions. We propose that due to the open-source access, as well as low cost and mass, the TickTag fills a technological gap in wildlife ecology and will open up new possibilities for wildlife research and conservation.

## Introduction

Satellite navigation technology, including the U.S.-owned Global Positioning System (GPS), has profoundly revolutionised the field of wildlife ecology [1–4]. To date, GPS loggers have been used to study the behaviour of a vast variety of species, allowing scientists to analyse movement with previously impossible spatiotemporal accuracy to within metres or milliseconds, even in remote areas [5]. With the ongoing miniaturization of electronics [2], the tagging of smaller and smaller species becomes feasible. There is a widely followed rule that the mass of tracking devices should not exceed 5% of the animals' body mass, to minimise negative effects on behaviour or survival [6,7]. However, as many species are small, this much discussed

**Funding:** The author(s) received no specific funding for this work.

**Competing interests:** The authors have declared that no competing interests exist.

5% limit has been occasionally exceeded [8,9]. While this does not seem to cause obvious negative effects in animals such as insects or when tags are only fitted for a short amount of time during non-demanding times in an animals' life, relative device mass is one of the most critical parameters in animal tracking [10–14]. Technological advancements providing high-definition tags with small mass may thus allow essential studies of small species with strong conservation concern and reduce the device impact on larger species.

Most of the currently available ultra-light tracking devices below one gram do not determine positions based on GPS, but transmit high-frequency radio signals [15–18] or use light sensors [19]. These technologies require much less energy and can operate on small batteries. However, they have large average location errors (e.g., 200 m—10 km for the ARGOS system [20,21] or 134 m for drone-based VHF tracking [22]). They also require a large amount of effort for localisation via handheld receivers or a geographically limited infrastructure at the study site (e.g., a network of Bluetooth ground nodes [17] or automated radio-telemetry arrays for the Motus Wildlife Tracking System [18]). To the best of our knowledge, the lightest currently available GPS loggers (including battery and housing) are the Technosmart Gipsy 6 at 1.5 g, the Lotek PinPoint 10 at 1.0 g and the Pathtrack nanoFix GEO Mini at 0.95 g [23–25]. These and similar miniaturized devices have been instrumental in understanding for example the territory usage of small migratory songbirds [26], but they all record a maximum of 200 positions on a single battery charge and remain proprietary and expensive [27]. As wildlife research often operates on limited budgets and or needs a large number of tracking tags, this results in many studies being done with only a handful of devices. This in turn limits how much data can be collected simultaneously and the conclusions about the animals' movement and ecology that can be drawn from them [27,28].

A different approach is to implement Snapshot (or Fastloc) GPS, where latitudes and longitudes are not calculated on-board, but reconstructed after data retrieval, based on milliseconds-long raw satellite data [29–33]. This method drastically reduces the energy usage per GPS fix. A recent prototype demonstrated that such devices can be powered for nearly two years by a 3 g CR2032 coin cell when recording at quarter-hourly intervals [30]. However, Snapshot GPS loggers normally produce less accurate positions [31], cannot react in situ to positional events (as required for geo-fencing and real-time monitoring [34]), and need large onboard memories for storing raw satellite data.

Open-source and do-it-yourself (DIY) designs of animal tracking devices are becoming increasingly popular, driven by the modularity and user-friendliness of the Arduino development platform [35–38]. Producing such devices is less expensive than buying commercial ones, but all of them are too large for small species below 50 g as they typically incorporate multiple combined circuit boards. For example, the open-source TNG logger weighs 15 g excluding battery [35,38]. The required skills and time-consuming hand assembly of electronic modules further limits the suitability of many of these devices. Thus, it is critical that a new practical and integrated solution is developed to fill an important gap in animal tracking. Ultra-light, low-cost GPS logging will make ethically sound field studies with increased sample sizes possible even with limited funding.

Here, we present the TickTag: a minimalistic open-source design for a reusable 0.65 g GPS logger that is operated via a simple smartphone-compatible user interface. The prototype is optimised for low-cost automatable production and for collecting the maximum number of fixes on even the smallest of batteries. The main focus when developing our system was to set new mass and performance standards for open-source GPS loggers. We field-tested our tag design on two dogs (*Canis lupus familiaris*), where the weight of the deployed logger corresponded to an average dog treat (0.01% of the body weight) and on ten greater mouse-eared bats (*Myotis myotis*), where previous GPS-logging exceeded the 5% limit [10].

The entire production-ready design (hardware, software, user interface, 3D-printable housing, assembly instructions) is provided open-source, and maintained in a publicly available repository: https://github.com/trichl/TickTagOpenSource or through Zenodo [39].

## Methods

### Hardware design

The TickTag hardware (Fig 1) consists of a 2-layered 25 x 10 x 0.15 mm flexible printed circuit board (Fig 1A) with only 16 surface-mounted components: an Atmel ATTINY1626 microcontroller with internal oscillators (Fig 1B), a Quectel L70B-M39 GPS module (Fig 1C), a Johanson Technology 1575AT43A0040E chip antenna (Fig 1E), an ON Semiconductor CAT24M01HU5I-GT3 128 kByte EEPROM memory (Fig 1F), a Texas Instruments TPS22916 load switch (Fig 1D), a Panasonic AXE610124 10-pin connector (Fig 1H), a green LED (Fig 1G) and nine passive components (resistors and capacitors). We chose components with wide operating voltage ranges between 3.0 and 4.2 V, to refrain from using any on-board voltage transformations (e.g., LDO regulators or DC converters), which benefits mass and power efficiency. The microcontroller orchestrates power, communication, individual tag configurations, and persistent data storage on the EEPROM memory (Fig 1F). Although smaller and lighter GPS modules exist (e.g., the 4 x 4 x 0.55 mm U-Blox M10 [40]), we chose the Quectel L70B-M39 (Fig 1C) due to its low cost (≤ 10 USD) and exceptionally low power consumption.

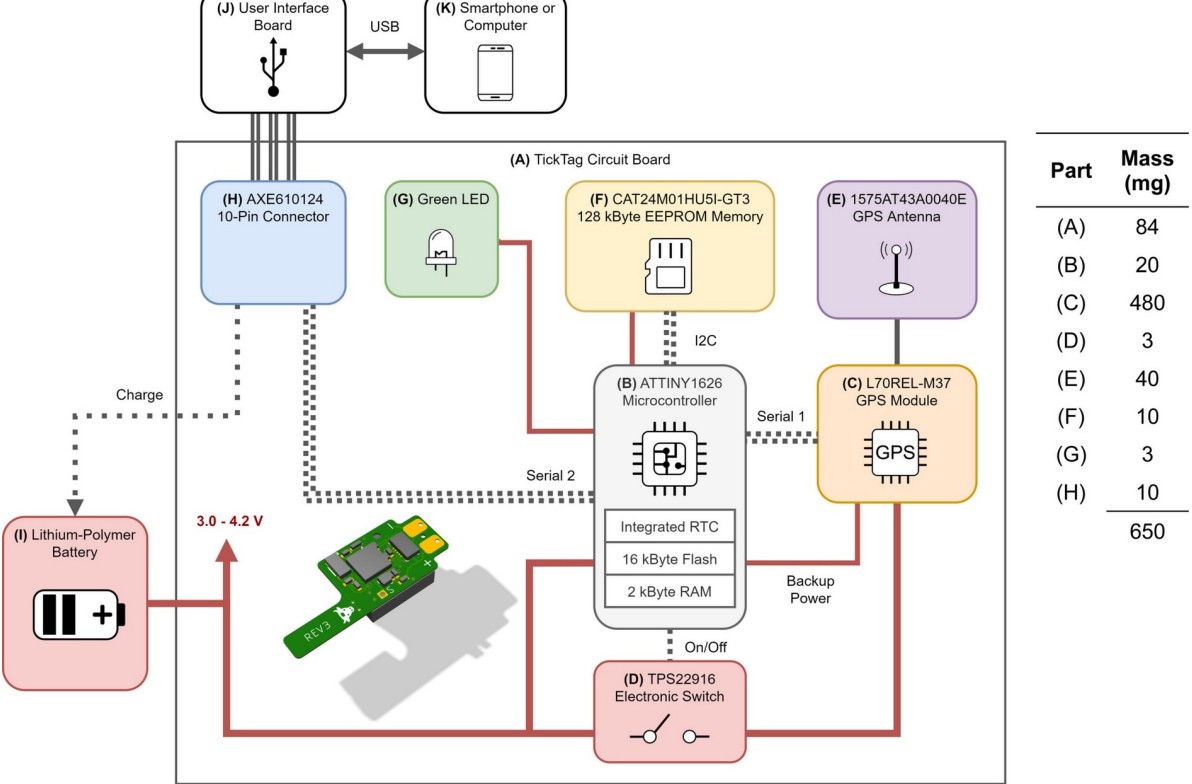

| Part | Mass (mg) |
|------|-----------|
| (A) | 84 |
| (B) | 20 |
| (C) | 480 |
| (D) | 3 |
| (E) | 40 |
| (F) | 10 |
| (G) | 3 |
| (H) | 10 |
| | 650 |

**Fig 1. Low power hardware design of the TickTag.** The microcontroller (B) manages the power states of the GPS module (C) via an electronic load switch (D) and a separately controlled backup power line. The EEPROM memory (F) is powered by a microcontroller pin and therewith requires energy only when reading or writing data. A green LED (G) gives visual feedback to the user (e.g., when the TickTag becomes activated). Data and configuration are accessible via a custom-designed user interface board (J) that can be connected to a computer or smartphone via USB (K).

We power the loggers with commonly used lithium-polymer batteries. These are limited by a size-dependant maximum continuous discharge current (e.g., 15 mA for a 30 mAh lithium-polymer battery), which for single cells below 1 g becomes too low to power most GPS receivers without degrading the battery capacity or introducing strong voltage drops [41,42]. The low power demand of the Quectel L70B-M39 significantly reduces the overall tag mass because it allows the use of batteries with capacities below 30 mAh.

## Embedded software design

We implemented the microcontroller software in Atmel Studio 7.0, using C as programming language. The software consists of a state machine controlling the data logging and a serial-based user interface. We optimised the code to fit on the size-limited 16 kByte flash memory of the ATTINY1626 and programmed the microcontroller via its 1-pin Unified Programming and Debugging Interface (UPDI) on an Arduino Nano with jtag2updi firmware [43]. We developed an on-board lossless compression algorithm that stores GPS fixes (UTC timestamp, latitude, longitude) as consecutive 10-byte datasets (see Github repository for details [39]). This enables the TickTag to persistently store up to 13,100 GPS fixes on the 128 kByte EEPROM memory. The TickTag software automatically generates statistics for assessing the GPS performance, including average horizontal dilution of precision (HDOP), time to first fix, average time to fix and fix success rate. The ATTINY1626 can read its own supply voltage and is programmed to enter hibernation mode when the battery voltage falls below a user-configurable threshold (e.g., 3.3 V). This makes additional protection circuits on the lithium-polymer battery unnecessary and further reduces tag mass.

## Software strategies for GPS logging on animals

We implemented several measures in software to optimise the number of achievable GPS fixes when using small batteries. The microcontroller is clocked down to 1 MHz (700 µA when active), which is the minimum frequency for serial communication with the GPS module at 9600 baud. Fix attempts are stopped after 120 s if the GPS module has not been able to digitise the current UTC time from a satellite signal by then, or after 300 s if the current UTC time is received, but no position could be estimated (e.g., due to an insufficient number of satellites in sight). After a failed fix attempt the system hibernates for 15 min, regardless of the configured GPS sampling interval. To reduce the location error, at the cost of slightly increased power consumption, the TickTag waits a maximum of 9 s for the HDOP to reach a user-configurable minimum value before storing the position estimate. Furthermore, users can configure a daily recording time window with minute resolution (e.g., between 6:20 am and 10:15 am UTC). Outside the recording time window, the TickTag hibernates at minimum power consumption and estimates the current time only with its internal real time clock (RTC; accuracy: ± 3% at 25˚C). After getting the first position estimate of a day ('cold start'), the TickTag keeps the GPS module powered for an additional 30 s. During this time, the receiver collects more satellite data, which is recommended by the GPS module manufacturer to reduce the time to subsequent fixes. Between fixes, the GPS backup domain stays powered via a microcontroller output pin, allowing for 'hot starts' with significantly reduced fix times. When configured to sample GPS fixes between 1 and 5 s intervals, the TickTag activates a proprietary fitness low power (FLP) mode of the L70B-M39 GPS module and keeps the receiver fully powered between fixes.

We implemented a self-configuring geo-fencing mode, that can optionally be activated via the user interface. Here, the TickTag memorises the first GPS fix after activation as its home location. GPS fixes within a 300 m range of the home location are not stored, but instead

trigger a 10-min hibernation state. Geo-fencing saves battery and memory capacity when for example only excursions outside an animal's home range or territory are of interest [34].

Instead of recording a single GPS fix per configured time interval, the TickTag can be programmed to periodically record GPS bursts with configurable lengths. Such high-frequency GPS bursts, recorded at 1 Hz, can reveal fine-scale movement patterns (e.g., linear feature tracking of red foxes [44]) and increase the number of achievable GPS fixes with a proportionally low impact on battery life (e.g., 71% increased consumption when recording 5 consecutive fixes instead of a single fix at an average time to first fix of 7 s). To minimise the effect of positional inaccuracies on the 1 Hz burst, the TickTag can be configured to start recording the burst only when reaching a minimum HDOP value.

## User interface

We developed a portable 66 x 18 mm electronic circuit board as a user interface (Fig 1J). Tick-Tags can be connected to the user interface board (UIB) via a micro-sized 10-pin connector (Fig 1H). The UIB allows for the i) safe recharge of the TickTag with an on-board battery management system that is optimised for small lithium-polymer batteries (charge current: 15 mA), ii) programming of the microcontroller, iii) establishment of serial communication with a computer or smartphone via a USB cable (for data download and configuration (see Github repository for parameter details [39])). The UIB integrates a physical push button that is used for starting and stopping the TickTag in the field. The tag can also be configured to auto-start after a configurable delay. We significantly reduced weight of the part of the TickTag that is deployed on the animal by outsourcing the user-interface-related functionalities to the UIB (e.g., USB connector, UART bridge, re-charging circuit, buttons). We also developed an open-source Android smartphone app that can be used in the field for simple and fast data download and tag configuration. The app communicates with the UIB via a USB On-The-Go (OTG) phone adapter and is also part of our open-source Github repository [39].

## Results

### Automated production leads to significant cost reduction

We generated all files necessary for an automated production of the TickTag and UIB (i.e., PCB specification, Gerber files, pick and place files, bills of materials) and sent them to an industrial PCB manufacturer (PCBWay) where the circuit boards of 75 TickTags and 15 UIBs were manufactured and assembled. We received the electronic boards after a few weeks, programmed the microcontrollers, and soldered batteries on (in total 60 s per tag) (Fig 2A). We documented all construction steps for easy and fast reproducibility (see Github repository for details [39]). The overall production costs when ordering 75 tags (including PCB manufacturing, assembly, shipping to Germany, German import duties and taxes) amounted to 26 USD per tag and an additional 6 USD for a 30 mAh lithium-polymer battery each. The UIB production costs 19 USD per unit when ordering a batch of 15 assembled boards.

### Performance assessment

We verified the energy demands of the TickTag with an Otii Arc battery simulator. The Tick-Tag required on average 14.5 mA while acquiring a position (in FLP mode: 9.5 mA), 8 μA between hot start fixes, and less than 500 nA when inactive, which is comparable to the self-discharge rate of a 30 mAh lithium-polymer battery (~ 5% per month [45]).

We conducted outdoor tests with various configurations to evaluate commonly used performance metrics of GPS loggers [27,46–48], including HDOP, time to first fix, time to fix, fix

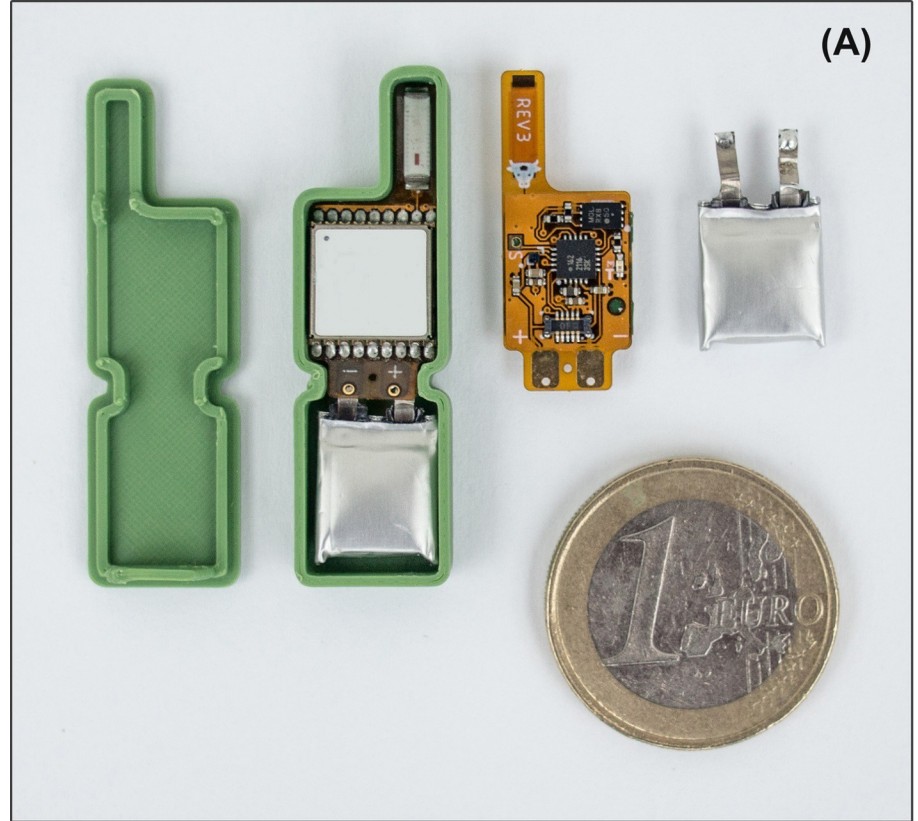
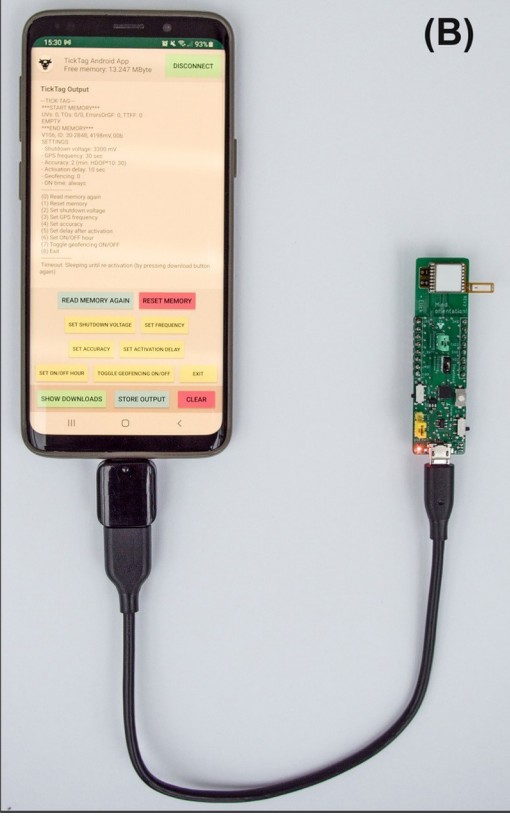

**Fig 2. TickTag including housing example (A) and user interface (B).** Here, the electronics (0.65 g) are powered by a 30 mAh lithium-polymer battery (0.55 g) and housed in a 3D-printed case (1.25 g), which can be replaced by Parafilm for weight reduction. The UIB can be connected to a computer or Android smartphone (via a USB OTG phone adapter) with the open-source TickTag app (B), allowing for tag configuration, data download and re-charging the battery.

success rate, location error and maximum runtime (Table 1). Fix success rates, average HDOPs and average fix times were calculated and stored onboard the TickTag. The device was positioned in a suburban area and remained stationary for the test trials. All tests were conducted in winter at low outside temperatures (-4 to +6˚C), where the performance of lithium-polymer batteries is known to be degraded [49]. We expect a higher number of average fixes in warmer environments than seen in our results here. TickTag location estimates were compared against the averaged locations of two GPS-enabled Android smartphones. We found a significant increase of achievable fixes per battery charge (up to 340 fixes per mAh (Fig 3A)) and accuracy (7.6 m average location error (Fig 3A and 3B)) when configuring the TickTag to sample high-frequency GPS data (0.2–1 Hz), which is due to the GPS module being fully powered between fixes and constantly receiving and updating satellite information. When configured to sample at more traditional intervals (30 s to 5 min), we measured an average fix success rate of 100%, and an average location error of 19.2 m. Burst recordings decreased the average time to fix by 4 s and the average location error by 7.2 m (37%), which is because the extended sampling periods improve predictions for following hot starts.

## Case studies on animals

In the first case study (Fig 4A), we equipped a 24 kg pet dog (*Canis lupus familiaris*) with a TickTag during six of its daily afternoon walks. The TickTag inside a 3D-printed housing was attached to the dog's harness with a 30 mAh lithium-polymer battery (total tag mass: 2.45 g,

**Table 1. Outdoor performance of the TickTag at a stationary position in a suburban area, operating on a 30 mAh lithium-polymer battery (total tag mass without housing: 1.2 g).**

| TickTag mode | GPS logging interval | Number of tests | Avg. fixes | Avg. time to first fix (s) (*) | Avg. time to fix (s) (**) | Avg. HDOP (***) | Avg. location error (m) (****) | Avg. fix success rate (%) | Avg. runtime (hrs) (*****) | Avg. current (mA) (******) |
|---|---|---|---|---|---|---|---|---|---|---|
| GPS stays on (fitness low power mode activated) | 1 s | 4 | 10,190 | 64 | - | 0.9 | 4.1 | 100 | 2.8 | 10.6 |
| | 5 s | 2 | 2,087 | 224 | - | 1.0 | 11.2 | 100 | 2.9 | 10.3 |
| Single fix mode | 30 s | 2 | 496 | 92 | 10 | 2.1 | 20.3 | 100 | 7.2 | 4.3 |
| | 1 min | 2 | 423 | 103 | 10 | 1.8 | 18.2 | 100 | 9.2 | 3.4 |
| | 5 min | 8 | 191 | 87 | 12 | 1.7 | 19.2 | 100 | 18.4 | 1.8 |
| Periodic 1 Hz burst mode | 10 s burst every 30 s | 2 | 5,597 | 118 | 7 | 1.5 | 12.3 | 100 | 7.4 | 4.1 |
| | 20 s burst every 2 min | 3 | 3,567 | 93 | 7 | 1.3 | 11.8 | 100 | 8.8 | 3.6 |

The tag was flat-oriented, with the antenna facing up to the sky, and attached to a piece of felt made from sheep wool to mimic a field deployment. Outside temperature was between -4 and +6°C. Each setting was tested multiple times on different days.

(*) Time required for obtaining a fix when no previous satellite information is available (cold start, e.g., after a longer sleeping period),

(**) Included both the initial cold start time and subsequent hot start times (where previous satellite information was available),

(***) The TickTag was configured to try achieving a HDOP of 3.0 or lower within 9 s after getting the first GPS-generated location estimate,

(****) Determined by using the Haversine formula for the great-circle distance between the GPS-generated location and the averaged locations of two GPS-enabled Android smartphones,

(*****) Can be distributed over several days when configuring daily recording windows (with minimal performance loss due to additional daily cold starts),

(******) Average tracking current while discharging the 30 mAh lithium-polymer battery from 4.2 V to 3.3 V.

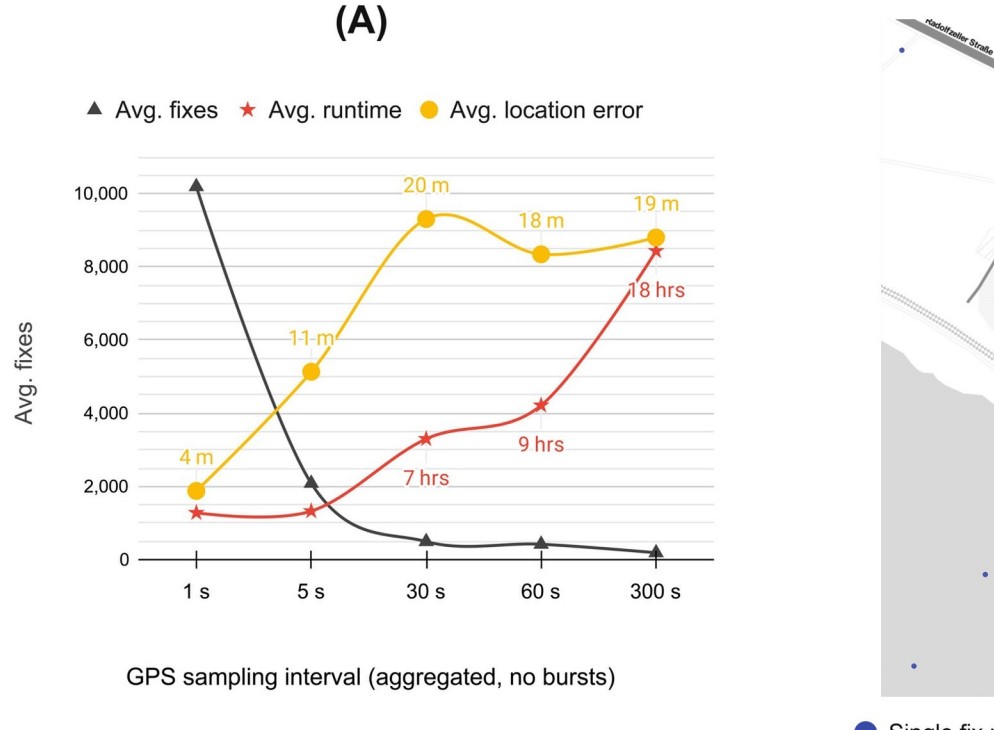

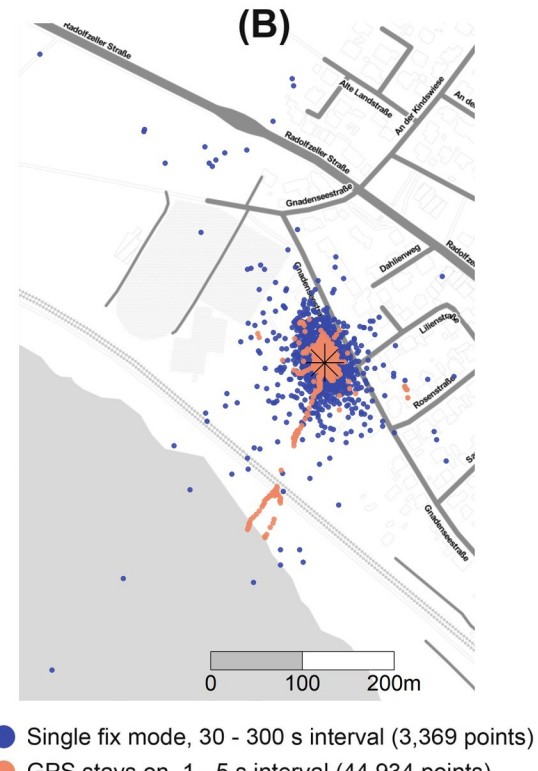

**Fig 3. TickTag performance comparison of GPS sampling intervals at a stationary position in a suburban area.** Map data from OpenStreetMap (OpenStreetMap contributors, http://www.openstreetmap.org/copyright). Map tiles by Stamen Design, under CC BY 3.0.

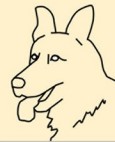

**(A)** Pet dog / anti-poaching hound (fix attempt every 1 s / 10 s fix bursts every 15 s)

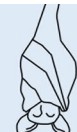

**(B)** Bat (fix attempt every 30 / 60 s)

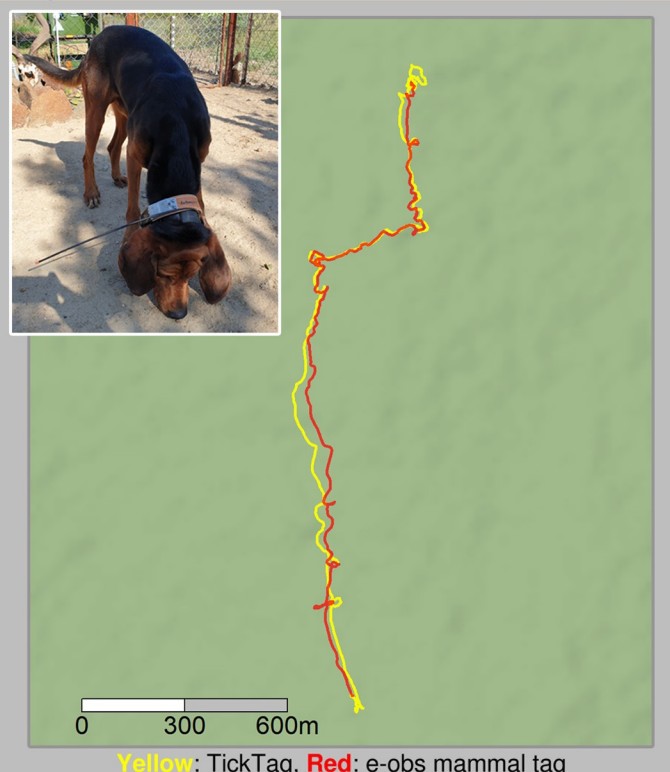

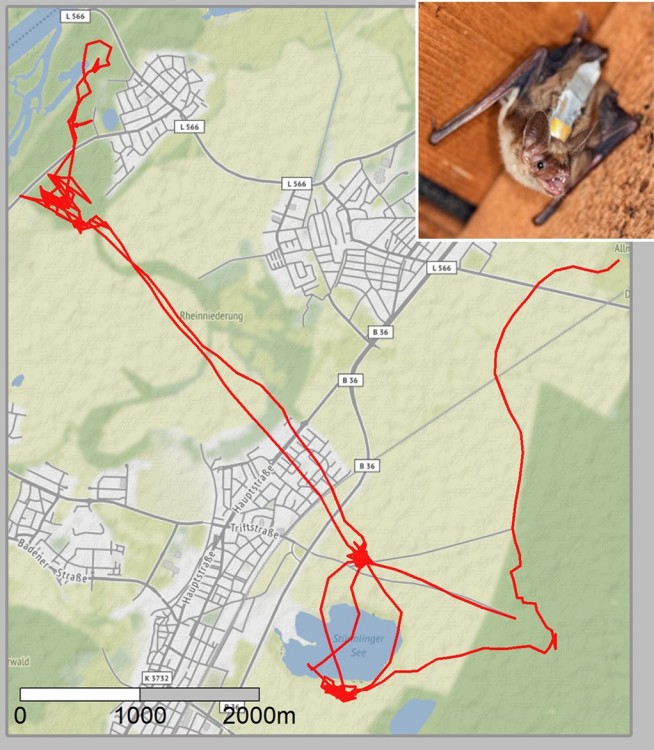

**Yellow**: TickTag, **Red**: e-obs mammal tag

| Part | Mass (g) |
|---|---|
| TickTag electronics | 0.65 |
| 30 mAh LiPo battery | 0.55 |
| 3D-printed housing | 1.25 |
| On animal | **2.45** |

| Part | Mass (g) |
|---|---|
| TickTag electronics | 0.65 |
| 30 mAh LiPo battery | 0.55 |
| Parafilm | 0.1 |
| On animal | **1.3** |

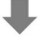

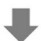

| Results | 1 Hz | 10 s burst |
|---|---|---|
| Avg. fixes per charge | 11,953 | 7,287 |
| Avg. fix success rate (%) | 100 | 100 |
| Avg. HDOP | 0.9 | 1.4 |
| Avg. runtime (hrs) | 3.3 | 6.3 |

| Results | 30 / 60 s |
|---|---|
| Avg. fixes per charge | 188 |
| Avg. fix success rate (%) | 90 |
| Avg. HDOP | 2.2 |
| Avg. runtime (hrs) | 2.9 |

**Fig 4. Evaluation of short-term case study deployments of the TickTag on dogs (A) and on greater mouse-eared bats (B).** We compared the TickTag GPS data of the anti-poaching hound with an e-obs mammal tag that was attached to the same collar (A). Map data from OpenStreetMap (OpenStreetMap contributors, http://www.openstreetmap.org/copyright). Map tiles by Stamen Design, under CC BY 3.0.

0.01% of the dog's body weight) and configured to record GPS data at 1 Hz, either continuously or in 10 s long bursts every 15 s. The pet dog was accustomed to wearing a harness. Owner consent was obtained prior to the tagging. The case study was conducted at Lake Constance, Germany. In order to test the GPS performance in various environments, each walk included a section of dense forest, hilly terrain, open grass plains and suburban area. After each one hr walk, we placed the TickTag at a stationary position and waited until the tag stopped recording due to low battery voltage (< 3.3 V). We connected the UIB to an Android phone to download data and recharge the TickTag between walks (Fig 2B). On a single 30 mAh battery charge, the TickTag recorded on average 11,953 fixes (in 3.3 hrs) with an average HDOP of 0.9 when sampling at 1 Hz. This corresponds to 398 fixes per mAh battery capacity, or 4,879 fixes per g overall device mass. In burst mode (10 s bursts every 15 s) the tag recorded on average 7,287 fixes (in 6.3 hrs) with an average HDOP of 1.4. We further attached a TickTag to the collar of an anti-poaching hound near Kruger national park, South Africa, as part of a larger behavioural experiment (total tag mass: 2.45 g, 0.01% of the dog's body weight) (Fig 4A). The field site was chosen due to its challenging natural environment, including high ambient temperatures (> 40°C) and thick bushland. Owner consent was obtained prior to sampling and the hound was accustomed to wearing a collar. Here, the TickTag recorded 1 Hz GPS during a 20 min long training exercise, where a pack of hounds was following human tracks in the bush. We attached an e-obs mammal tag [50], also capable of recording 1 Hz GPS but in a 250 g unit, to the same collar and compared the 1 Hz location estimates to the TickTag results. The location difference between the TickTag and the e-obs mammal tag was on average 67 m (calculated by using the Haversine formula). The average HDOP was 1.1 for the TickTag locations and 1.6 for the e-obs tag locations. A lower HDOP value is an indicator for higher precision, but does not guarantee a lower location error, so we could not conclude which track was more accurate [5,51].

In the second case study (Fig 4B), we deployed TickTags on ten greater mouse-eared bats (*Myotis myotis*) that were captured inside their roost, a roof truss in a suburban area, near Karlsruhe, Germany. Electronics and 30 mAh batteries were wrapped in Parafilm for weather protection, then glued to the back of the bats with Osto-Bond skin bonding cement (Montreal Ostomy & Home) and removed from the animals on the following day (total tag mass: 1.3 g, ≤ 5% of the bat body weights). Tags were configured to record GPS positions between 10:40 pm and 7:00 am (GMT+2), with different interval settings (30 s, 60 s). Inside the roost site GPS signals were blocked by the building's roof structure. We installed an indoor GPS repeater to avoid energy-costly GPS acquisition timeouts on the tags. We activated the geo-fencing mode, to save battery power while the tagged bats remained in the colony. Due to malfunctioning lithium-polymer batteries, six devices did not record data. The remaining TickTags recorded on average 188 fixes (6.3 fixes per mAh battery capacity, or 144 fixes per g device mass) and at best 410 fixes (13.7 fixes per mAh battery capacity, or 315 fixes per g device mass). The collected data revealed previously unknown local hunting grounds and travel routes of the bats in a suburban area. All authors complied with the legislation in the respective countries where fieldwork was conducted. Data from bats were collected under ethical permission from Regierungspräsidium Karlsruhe Az. 8852.15 Fledermäuse Dietz_Test Sender MPI. Because our first case study was conducted on a species not currently listed as threatened or endangered and the added device mass was only 0.01% of the dog's body weight, additional permissions were not required.

## Discussion

Since 1991 GPS technology has revolutionised our understanding of animal movements, but its use is limited by the additional mass that can be added to animals as well as prohibitive cost

of devices [3]. The on-going miniaturisation of loggers can reduce negative effects on tagged animals and opens new opportunities for studying the species-rich body mass classes below 50 g, where current research data are sparse [2]. We developed the to date lightest open-source GPS-enabled logger for animal tracking (electronics mass: 0.65 g), that utilises high-frequency recordings in combination with low power hardware optimisations and software strategies (e.g., burst recordings) to achieve up to 10,190 fixes on a 30 mAh battery (on-animal weight with short-term weather protection: 1.3 g). We demonstrate that automated production of open-source electronics leads to a reduction of both tag price and manual preparatory work, allowing for field studies to scale-up more easily, even with limited financial resources. With our easy-to-use and Smartphone-compatible user interface, TickTags can be recharged, reused, and flexibly configured for case studies. Testing on animals in the field confirmed the efficient power usage of the tags. We encourage the open-source usage of the TickTag by providing hardware production files, software, user interface, 3D-printable housings and assembly instructions: https://github.com/trichl/TickTagOpenSource or through Zenodo [39].

In the weight class below 2 g only a few commercial GPS loggers exist, with prices ranging from 500 to 800 USD. The Lotek PinPoint 10 achieves on average 130 fixes per g device mass at a minimum total tag mass of 1 g [24] and has been used to identify non-breeding territories of small migratory songbirds [26]. The Technosmart Gipsy 5 has been deployed on echolocating bats to study their foraging and navigation behaviour [52,53]. The successor (Technosmart Gipsy 6) weighs at minimum 1.5 g with battery and backpack [25]. The Pathtrack nanoFix GEO Mini has been used to track Mediterranean storm petrels [54] and achieves on average 168 fixes per g device mass at a minimum total tag mass of 0.95 g [23]. In comparison, the TickTag records up to 7,838 fixes per g device mass and is the first device in this weight class to introduce short-term high-definition GPS logging, at much reduced cost per tag (currently 32 USD including rechargeable battery). Our design integrates a 7 x 2 x 0.8 mm chip antenna instead of a commonly used 40 to 50 mm long quarter-wave whip antenna, to reduce the impact on tagged animals. To the best of our knowledge, there is no open-source GPS logger that is comparable to the TickTag regarding device mass or production price. The open-source TWLogger can be built for 90 USD and weighs 18.7 g without battery and housing [55]. The TNG logger costs 40 USD and weighs about 15 g without battery and housing [35,38]. Despite the size of the TickTag, our stationary evaluations and case studies show that the average time to fix (13 s), average fix success rate ($\geq$ 90%) and average location error (13.5 m) of the Tick-Tag are comparable to other, much heavier GPS loggers used for animal tracking (*cf.*, [27]), even in forests and bushlands. We find that burst and continuous 1 Hz recordings significantly improve the performance when compared to single fixes (-49% average location error, -34% average time to fix, +1,349% average fixes per mAh).

Our case study on dogs demonstrates how micro-sized GPS loggers can reduce the tagging impact on larger species, down to 'practically' zero (0.01% of the body weight), while still recording conclusive short-term fine-scale GPS data (up to 11,953 fixes on a 30 mAh battery charge). Our case study on greater mouse-eared bats (*Myotis myotis*) shows that the TickTag can be deployed on small free-ranging animals, without exceeding 5% of the body mass, a limit that was exceeded in previous GPS-tracking studies of this species [10]. Low weight solutions like the TickTag also open new attachment possibilities (e.g., necklaces instead of backpacks).

We emphasise that the system is most appropriate for short-term experiments where researchers can recapture animals for data download. The tag performance is best when sampling at high GPS frequencies. The TickTag persistently stores a maximum of 13,100 fixes (which equals to 3.6 hrs at 1 Hz, 18.2 hrs at 0.2 Hz or 14.6 hrs at 10 s long 1 Hz bursts every 30 s). This storage limitation needs to be considered when choosing batteries with higher

capacities (i.e., a TickTag attached to a 150 mAh battery and configured to record data at 1 Hz fills the memory before the battery is empty). The maximum recording frequency of the Tick-Tag is 1 Hz and the current design does not incorporate wireless transmission of data (such as GSM, LoRa, Sigfox, Bluetooth Low Energy, or WiFi [56–60]). Such a communication function could be implemented in the future but would increase electronics mass and power consumption. Although smaller GPS chips already exist (e.g., the 4 x 4 x 0.55 mm U-Blox M10 [40]), they need two or three times more power than the 0.48 g GPS chip we integrated. The electronics mass of the TickTag could further be reduced when a next generation of miniaturized ultra-low-power and low-cost GPS modules becomes available on the market.

To reduce negative impacts on study organisms and to collect genuine scientific data, the mass of animal-borne loggers should be minimised as much as possible [2,8,9,61,62]. Encouraging ecologists to deploy smaller devices, even on larger species, calls for a high-performing technological solution that provides data with a similar quantity and quality as larger devices, but at the same time does not increase expenses. With the TickTag we provide a valuable and affordable tool for the bio-logging community and our tests on animals demonstrate the remarkable compromise between extremely light weight and low production cost of the GPS loggers. Researchers can use our open-source design as it is, or develop it further (e.g., incorporating additional sensors or wireless transmission). The TickTag will be maintained regularly on the Github site, which allows for user contributions.

## Supporting information

**S1 File.**
(DOCX)

## Acknowledgments

We are grateful to the representatives of the Regierungspräsidium Karlsruhe for their support and discussions on tagging bats and especially to Kerstin Bach for her help during fieldwork and ideas for improving the tags and tagging. We would like to thank the Landwirtschaftliches Technologiezentrum Augustenberg, Außenstelle Forchheim for getting access to the colony and the opportunity to study the bats. Further, we thank Ramona Schniedermeier for the animal illustrations, Stephen Tyndel for the valuable comments on the manuscript and the engineering team at the Max Planck Institute of Animal Behavior for their discussions and advice in the development of the TickTag. Special thanks to the Southern African Wildlife College, including Johan van Straaten, as well as Louis van Schalkwyk and Pauli Viljoen from Contemplate Wild for their field support in South Africa. All authors complied with the legislation in the respective countries where fieldwork was conducted. Data from bats were collected by Christian Dietz under ethical permission from Regierungspräsidium Karlsruhe Az. 8852.15 Fledermäuse Dietz_Test Sender MPI.

## Author Contributions

**Conceptualization:** Timm A. Wild, Jens C. Koblitz, Dina K. N. Dechmann, Mirko Meboldt, Martin Wikelski.

**Data curation:** Timm A. Wild.

**Formal analysis:** Timm A. Wild.

**Funding acquisition:** Martin Wikelski.

**Investigation:** Timm A. Wild.

**Methodology:** Timm A. Wild, Christian Dietz, Martin Wikelski.

**Project administration:** Martin Wikelski.

**Resources:** Timm A. Wild, Christian Dietz.

**Software:** Timm A. Wild.

**Supervision:** Mirko Meboldt, Martin Wikelski.

**Validation:** Timm A. Wild, Jens C. Koblitz, Christian Dietz.

**Visualization:** Timm A. Wild.

**Writing – original draft:** Timm A. Wild.

**Writing – review & editing:** Dina K. N. Dechmann, Christian Dietz, Martin Wikelski.

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
