## [Decision Letter · Decision Letter 0]

14 Apr 2022

Micro-sized open-source and low-cost GPS loggers below 1 g minimise the impact on animals while collecting thousands of fixes

PONE-D-22-08082

Dear Dr. Wild,

We’re pleased to inform you that your manuscript has been judged scientifically suitable for publication and will be formally accepted for publication once it meets all outstanding technical requirements.

Kind regards,

Giorgio F Gilestro, PhD

Academic Editor

PLOS ONE

Additional Editor Comments (optional):

Reviewers' comments:

Reviewer's Responses to Questions

**Comments to the Author**

1. Is the manuscript technically sound, and do the data support the conclusions?

Reviewer #1: Yes

2. Has the statistical analysis been performed appropriately and rigorously? 

Reviewer #1: I Don't Know

3. Have the authors made all data underlying the findings in their manuscript fully available?

Reviewer #1: No

4. Is the manuscript presented in an intelligible fashion and written in standard English?

Reviewer #1: Yes

5. Review Comments to the Author

Reviewer #1: Excellent project, well documented.

I have worked with the Alexei Vyssotskiand "Neurologgers" for EEG recording in rodent studies, as well as designing data-logger hardware for similar applications. I am therefore familiar with the issues of weight, size, and battery life - as well as the vagaries of actual deployment and challenges thereof to produce robust systems that can work reliably in the field, not just under ideal conditions on a workbench.

These challenges are sufficiently described in the text to show the reasoning for making the various design/implementation choices, e.g. the tradeoffs between different GPS operating modes.

In addition to the hardware design, the authors have paid particular attention to the data recovery and battery recharging aspects of the project which is a very important part of having a successful setup that will work reliably in use.

This paper describes the design and implementation in good detail, I am confident that I could build these devices myself from the supplied information in the paper and the GitHub repository.

Two points in the above review questions - which should NOT be taken as a negative for this paper.

Point #2 "statistical analysis" - My background is hardware and microcontroller design, not mathematics. The described analysis seems sound to me, but I am not sufficiently experienced in this level of analysis to be able to decisively state if this would be the preferred method to be used.

Point #3 "all data" - I can't see the raw-data for the various tests described. I assume this will be presented in the supporting information or added to the GitHub repository as examples of use?

6. PLOS authors have the option to publish the peer review history of their article (what does this mean?). If published, this will include your full peer review and any attached files.

Reviewer #1: **Yes: **Susan Parker

---

## [Editor Report · Acceptance letter]

24 May 2022

PONE-D-22-08082 

Micro-sized open-source and low-cost GPS loggers below 1 g minimise the impact on animals while collecting thousands of fixes 

Dear Dr. Wild:

I'm pleased to inform you that your manuscript has been deemed suitable for publication in PLOS ONE. Congratulations! Your manuscript is now with our production department. 

Kind regards, 

on behalf of

Dr. Giorgio F Gilestro 

Academic Editor

PLOS ONE